Eutrophication trends in the coastal region of the Great Tokyo area based on long-term trends of Secchi depth

Akada Hideyuki 1
http://orcid.org/0000-0001-8031-7881 Kodama Taketoshi 2 takekodama@g.ecc.u-tokyo.ac.jp
Yamaguchi Tamaha 3
1 Kanagawa Prefectural Fisheries Technology Center , Miura, Kanagawa , Japan
2 Graduate School of Agricultural and Life Sciences, University of Tokyo , Tokyo , Japan
3 Fisheries Resources Institute, Fisheries Research and Education Agency , Yokohama, Kanagawa , Japan
Kuwae Tomohiro
Electronic publication date: 2023 Jul 28
Publication date: 2023
Volume: 11
Electronic Location ID: e15764
Received 2023 May 4; Accepted 2023 Jun 27
Copyright: © 2023 Akada et al.
Copyright year: 2023
Copyright holder: Akada et al.
License: This is an open access article distributed under the terms of the Creative Commons Attribution License, which permits unrestricted use, distribution, reproduction and adaptation in any medium and for any purpose provided that it is properly attributed. For attribution, the original author(s), title, publication source (PeerJ) and either DOI or URL of the article must be cited.
License URL: https://creativecommons.org/licenses/by/4.0/

Keywords: Coastal manegement, Water quality, Oceanography, Coastal oceanography, Ocean optics, Ocean color, Phytoplankton, Primary production, Onboard observation

Funding: JSPS KAKENHI 19K06198, 23H02285 Research and Assessment Program for Fisheries Resources Fisheries Agency of Japan Japan Fisheries Research and Education Agency This work was supported by JSPS KAKENHI Grant Numbers (19K06198, 23H02285), the Research and Assessment Program for Fisheries Resources, the Fisheries Agency of Japan, and a general grant from the Japan Fisheries Research and Education Agency. The funders had no role in study design, data collection and analysis, decision to publish, or preparation of the manuscript.

==============================
Background

The coastal ocean’s environment has changed owing to human activity, with eutrophication becoming a global concern. However, oligotrophication occurs locally and decreases fish production. Historically, the Secchi depth has been used as an index of primary productivity. We analyzed the results of over-a-half-century routine observations conducted in Sagami Bay and Tokyo Bay to verify the eutrophication/oligotrophication trend based on Secchi depth observations in a temperate coastal region near the Greater Tokyo area, which is highly affected by human activities.

Methods

Data recorded in the Kanagawa Prefecture from 1963 to 2018 were used in this study. After quality control, the observation area was divided into Tokyo Bay, the Uraga Channel (outer part of Tokyo Bay), Sagami Bay (northern part), and Sagami Nada (southern part of Sagami Bay) based on temperature and salinity at a depth of 10 m. Because the environmental parameters showed autocorrelation, time-series and correlation analyses were conducted using generalized least squares (GLS) models with a Prais-Winsten estimator.

Results

The Secchi depth was the shallowest in Tokyo Bay, followed by the Uraga Channel, Sagami Bay, and Sagami Nada, and was deep in winter (December and January), and shallow in summer (July) in all regions. The correlated analyses using the GLS model indicated that the shallowing of Secchi depth was significantly associated with decreases in temperature, salinity, and phosphate concentration. However, time-series analyses using GLS models indicated that the Secchi depth was significantly shallower, except in Tokyo Bay, where the surface temperature was significantly warming and the surface phosphate and nitrite concentrations decreased everywhere. A significant shallowing trend of the Secchi depth was mostly observed during the light-limiting season (January–March).

Discussion

Correlation analyses suggested the importance of horizontal advective transport, particularly from Tokyo Bay, which has cold and less saline eutrophic water. However, long-term shallowing of the Secchi depth was associated with warming, and changes in salinity were not significant in most months when the Secchi depth trend was significant. Thus, horizontal advection is not the primary cause of long-term eutrophication. Because the eutrophication trend was primarily observed in winter, when light is the major limiting factor of primary production, we concluded that warming provides a better photoenvironment for phytoplankton growth and induces eutrophication. As a decline in anthropogenic nutrient input after 1990s was reported in the investigated area, the long-term eutrophication trend was most likely caused due to global warming, which is another alarming impact resulting from human activities.

Introduction

Sixty percent of the human population lives near coastal areas worldwide, and coastal oceans are strongly affected by anthropogenic activities (Vitousek et al., 1997). In the coastal ocean, eutrophication, which is defined as “an increase in the rate of supply of organic matter to an ecosystem” (Nixon, 1995), potentially affects the natural environment (Paerl et al., 2014). Both anthropogenic nitrogen and phosphorus inputs drive eutrophication, and various studies have focused on anthropogenic nitrogen inputs (Paerl et al., 2014). Anthropogenic nitrogen inputs to coastal oceans usually originate from rivers, and the utilization of synthetic fertilizers and legume agriculture increases the nitrogen supply to rivers and coastal oceans (Malone & Newton, 2020). Globally, nitrogen transported from rivers to oceans increased by 43% between 1970 and 2000 (Breitburg et al., 2018). Global warming and climate change are the major factors contributing to ever-changing coastal environments (Paerl et al., 2014; Robins et al., 2016; Sinha, Michalak & Balaji, 2017). The difference in rainfall patterns due to climate change affects nitrogen inputs to rivers (Sinha, Michalak & Balaji, 2017), and the warming of water allows for an increase in harmful organisms and a decrease in biodiversity (Robins et al., 2016).

Although eutrophication in coastal areas is a global concern, coastal management and warming water decrease productivity, named as “oligotrophication” (Yamamoto, 2003; Yanagi, 2015), in some coastal waters such as the Mediterranean Sea (Le Fur et al., 2019), the North Sea (Støttrup et al., 2017), Narragansett Bay, USA (Nixon et al., 2009), Tokyo Bay, Japan (Kubo et al., 2019), and Seto Inland Sea (Yamamoto, 2003; Yanagi, 2015). In the Mediterranean Sea, oligotrophication and warming trends have changed vegetation including phytoplankton assemblages (Derolez et al., 2020b; Le Fur et al., 2019), and shellfish farming (Derolez et al., 2020a). Oligotrophication due to coastal management and warming may lead to a decline in fishery production in the North Sea (Støttrup et al., 2017) and Narragansett Bay (Nixon et al., 2009), respectively. In Japanese coastal areas, oligotrophication is considered to be a decline in fishery production in the Seto Inland Sea (Abo & Yamamoto, 2019; Yamamoto, 2002, 2003), and to be caused by advanced wastewater treatment (Kubo et al., 2019; Kubo, Imaizumi & Yamauchi, 2020), whereas the effects of agriculture, as pointed out by Malone & Newton (2020), are ignored.

The Secchi depth, also known as transparency, is one of the key parameters that reveals long-term changes in primary productivity in the ocean, and it is widely used over a century (Boyce, Lewis & Worm, 2010). The Secchi depth is balanced by the phytoplankton biomass in the ocean; thus, an increase and decrease in Secchi depth are signatures of oligotrophication and eutrophication, respectively (Boyce, Lewis & Worm, 2010; Falkowski & Wilson, 1992; Lewis, Kuring & Yentsch, 1988). In the North Pacific, the chlorophyll a concentration converted from the Secchi depth decreased (Boyce, Lewis & Worm, 2010). Secchi disk observations are intensively conducted particularly in the waters near Japan (Boyce, Lewis & Worm, 2010; Falkowski & Wilson, 1992), and long-term variations in Secchi depth in these areas are frequently reported, including in documents written in Japanese (Hagiwara, 2009; Hayami, Maeda & Hamada, 2015; Hisamatsu et al., 2008; Ishida & Ueda, 2008; Ishii, 2009). Based on these area-specific results, a deepening in Secchi depth was observed in the Kii Channel in southwestern Japan (Ishida & Ueda, 2008) and in Ariake Bay, western Japan (Hayami, Maeda & Hamada, 2015), whereas no clear variations were observed in Kesennuma Bay, northeastern Japan (Hisamatsu et al., 2008), and Enshu Nada, central Japan (Hagiwara, 2009).

Tokyo and Sagami Bays are expected to have high anthropogenic impacts because pollution load originating from the Great Tokyo area, which has more than one-third of the Japanese population (43 million), flows into these bays. Tokyo Bay is a semi-closed hypereutrophic bay, whereas Sagami Bay is an open bay affected by the Kuroshio and is much more oligotrophic than that of Tokyo Bay (Ara et al., 2011; Ara & Hiromi, 2008). Because these bays are adjacent to each other, their waters are exchanged frequently (Aoki et al., 2022; Furushima, 1996). The trend of oligotrophication in Tokyo Bay has been previously reported (Ishii, 2009; Kubo et al., 2019). Kubo et al. (2019) indicated that advanced wastewater treatment is a possible reason. In addition to the artificial effects, Aoki et al. (2022) suggested that the weakening of the nutrient input from Sagami Bay contributes the oligotrophication. While long-term variation in oceanic environments has not been reported in Sagami Bay, fishery production using set nets has gradually decreased over the past five decades (Takamura, Katayama & Kinoshita, 2016). Based on the relationships between oligotrophication and the decline of fishery production in other areas (Abo & Yamamoto, 2019; Nixon et al., 2009; Støttrup et al., 2017), the decline of fish production in Sagami Bay may suggest oligotrophication therein. Hence, this study aimed to identify the trends in oligotrophication/eutrophication and the possible underlying mechanisms in Sagami and Tokyo Bays based on long-term monthly onboard ocean monitoring conducted in Kanagawa Prefecture.

Materials and Methods

Database of the ocean-going monthly observations

Monthly ocean-going observations have been conducted since April 1963 in Tokyo and Sagami Bays (Fig. 1), using vessels belonging to Kanagawa Prefectural Fisheries Technology Center. There were total 29,460 stations with unique ID based on the observation year, month, and geographic position. The data observed during the 20th century were primarily collected from field books or paper documents issued by Kanagawa Prefecture; resulting in some missing information such as equipment types. Observations of the target month were always conducted between the end of the preceding month and the beginning of the target month. For example, when the target month is February, the observation was always conducted between the 25th of January and the 10th of February.

Figure 1 Map showing sampling stations.

Points on the small-scale map (right) denote 70 sampling stations. Crosses, closed triangles, open circles, and closed squares denote the regularly observed 35 stations classified as Tokyo Bay, the Uraga Channel, Sagami Bay, and Sagami Nada, respectively, based on the clustering analysis (Fig. 2). Small closed circle indicates irregularly observed stations that were not used to evaluate long-term trends.

Figure 2 Dendrogram of hierarchical clustering analysis based on temperature and salinity at 10 m depth of the routinely observed 35 stations.

The superimposed map denotes the distribution of the 35 stations. The stations were divided in four clusters (I–IV), and their geographical distributions were associated with Sagami Bay, Sagami Nada, the Uraga Channel, and Tokyo Bay, denoted similarly as in Fig. 1.

Various items described in the documents included date; location; Forel-Ule scale water color; Secchi depth; and vertical profiles of temperature, salinity, nitrate, nitrite, ammonium, and phosphate concentrations. Only the year and target month were recorded as dates. Temperature and salinity were recently measured using a conductivity-temperature-salinity (CTD) sensor (SBE911plus, Sea-Bird Electronics, Inc., Bellevue, WA, USA); however, there was no information on when the CTD observations started and what equipment was used until then. Temperature and salinity were documented at the 14 depths (0, 10, 20, 30, 50, 75, 100, 150, 200, 250, 300, 400, 500, and 600 m). Samples for nutrient analysis were collected at 0, 10, 30, 50, 100, and 250 m. The sampling layer was unchanged when the bottom depth of the station was shallow. Water samples for nutrient analyses were kept frozen until on-land measurements during these 30 years, but details before then are not described. Nutrient concentrations were manually measured from April 1967 to June 2010, after April 2012, they were measured using an autoanalyzer (QuAAtro2-HR, BLTEC) based on standard methods. Nutrient data from July 2010 to March 2012 were unavailable.

Other database

The monthly variations of precipitation data at the Tokyo observatory (35°41.5′N, 139°45.0′E) were downloaded from Japan Meteorological Agency. The annual means of nitrate plus nitrite, total nitrogen (TN), and total phosphorus (TP) concentrations in the Sagami River (Mairi Bridge, at 35°19.9′N 139°22.0′E) which flows into Sagami Bay were observed from 1970, 1984 and 1980, respectively, and downloaded from the water information system, Ministry of Land, Infrastructure, Transport and Tourism (http://www1.river.go.jp/; accessed April 9th 2023). Those monthly concentrations could not be obtained. The Kuroshio axis position was provided from Marine Information Research Center, Japan Hydrographic Association. The Kuroshio axis position is the fee-charging content, and thus we did not show it as figures and did not put it in the data file.

Data quality check and trimming

Because the methodology for historical ocean-going observations was unknown, data quality checks and validations were conducted on our database. First, we removed data whose temperature was >30 °C or whose salinity was >36. When the data exceeded these thresholds, the value was one to two orders of magnitude higher than the usual value. Data with Forel-Ule scale watercolor ≥22 and the Secchi depth ≥100 m was also removed. A Forel-Ule scale watercolor of ≥12 was only observed in 1973 and 1983 before 1993, while Secchi disk depth was sometimes recorded ≤2 m before 1993. This indicates that Ule-scale observations were not frequently conducted before 1993; thus, the Forel-Ule scale watercolor data were not used for further trend analysis. While Ule-scale observations were partly limited, the relationship between Secchi depth and Forel-Ule scale water color was significantly negative (Fig. S1), indicating that the Secchi depth is a good indicator of ocean surface color. Surface chlorophyll a concentration was not measured in the monitoring program by Kanagawa Prefecture, but Shibata & Aruga (1982) reported that the surface chlorophyll a concentration showed a strong agreement with Secchi depth even in Tokyo Bay. Thus, the Secchi depth could be treated as the index of phytoplankton biomass in our study.

Regarding nutrient concentrations, variations in nitrate and ammonium concentrations indicated possible artificial errors. At a depth of 100 m, where nutrient samples were only collected in Sagami Bay after 1993, the yearly median values of nitrate concentration were usually ~10 µM; however, those in 2006 and 2007 were ~2 µM, and those in 2012 and 2013 were >40 µM. Forty-micromolar nitrate at a depth of 100 m has never been observed downstream of the Kuroshio and Sagami Bays (Aoki et al., 2022; Hashihama et al., 2008; Kodama et al., 2014b). The reliable ammonium measurement was not enough, but rarely >0.5 µM in this area (Kodama et al., 2014a; Yasunaka et al., 2018). However, the yearly median ammonium concentration at 100 m depth was usually >0.5 µM before 2010. Thus, the nitrate and ammonium concentrations during all periods were not used in the downstream analysis.

Using the same criterion, phosphate and nitrite concentrations were considered reliable because manual standard methods were simpler than those for nitrate and ammonium. For the phosphate concentration, further validation was conducted using its relationship with temperature. A negative correlation between phosphate and temperature was expected based on Kodama et al. (2014b), using Eq. (1):

(1) [PO4]∼a×T+b

where, [PO4] and T represent the phosphate concentration and temperature, respectively. We calculated the slope (a) and intercept (b) for each year using data at depths ≥20 m. The mean slope and intercept with standard deviation (SD) values were −0.0673 ± 0.185 µM/°C and 1.69 ± 3.36 µM, respectively. The outlier slope and intercept values, defined as those without mean ± 1 × SD, were observed in 1967–1969 and 2003, and thus the phosphate concentration in these years was not considered. No outliers were found in the nitrite data.

Observations were conducted at over 70 unique stations during this observation period, including stations observed for only 1 year (Fig. 1). The frequency of observations indicated that 35 stations have been repeatedly observed over the last 50 years (Fig. 1).

Division to subareas

At some of 35 stations showed similar changes of parameters, and the observational errors are not completely removed by the above quality check and trimming processes. Thus, we classified 35 stations into several subareas using Ward’s hierarchical clustering analysis with Euclidean distance of temperature and salinity at a depth of 10 m. Consequently, the 35 stations were classified in four clusters (Fig. 2), and the cluster distributions agreed with the geographic distributions (Sagami Bay, Sagami Nada, Uraga Channel, and Tokyo Bay). Cluster I, which contained 13 stations, was observed in Sagami Bay. Cluster II, containing 13 stations, was primarily observed in offshore area of Sagami Bay, named as “Sagami Nada.” Cluster III, which contained only three stations, was observed in offshore of Tokyo Bay, named as “the Uraga Channel.” Cluster IV, which contained six stations, was observed in the Inner Tokyo Bay. A downstream analysis was conducted based on the median for each subarea calculated based on routine 35 stations.

Statistical analysis for long-term trend

Using the median of each subarea and month, all statistical calculations were conducted using R (R Core Team, 2023). The significance of seasonal and long-term trends in Secchi depth, temperature, salinity, phosphate concentration, and nitrite concentration at 10 m depth were investigated using a linear model with a Prais-Winsten estimator. A Mann-Kendall analysis is usually used to detect a trend (Kubo et al., 2019), but this approach can only show significantly positive, significantly negative, or insignificant trends, and we could not determine the change rate. Therefore, we calculated the trend using a linear model with the Prais-Winsten estimator. These trend analyses were conducted for the medians of each month and subarea using Eqs. (2) and (3).

(2) Yi,m∼a×year+b

(3) Yi∼a×year+f(month)

where, Y, i, m, a and b denote the response variables (Secchi depth, temperature, salinity, phosphate, nitrite, precipitation amount at Tokyo), subarea, target month, coefficient and intercept, respectively. In Eq. (3), the f value indicates that the month was transformed into categorical values to allow for nonlinear seasonal variations as intercepts. Thus, the interannual trend was identified regardless of the month.

To identify the factors controlling the variation in Secchi depth, a linear model was developed using temperature, salinity, nitrite, phosphate, precipitation at Tokyo, Kuroshio axis position, and year as response variables. Before setting the model, seasonality was removed from all the data as follows: (1) the monthly median (monthly climatological values) was calculated for each area and parameter and (2) the difference from the monthly climatological values was calculated. These differences were not seasonal (analysis of variance (ANOVA): df = 11, p > 0.05). Most of these differences were autocorrelated (Durbin-Watson test, threshold p = 0.05); autocorrelation was not identified for the Secchi depth in Tokyo Bay (n = 539, DW = 1.8652, p = 0.05355) or phosphate concentration in Sagami Nada (n = 527, DW = 1.9102, p = 0.1408). Thus, the Prais-Winsten estimator was used in the following linear model (Eq. (4)): (4) ΔZSD∼a1×ΔT+a2×ΔS+a3×ΔNO2+a4×ΔPO4+a5×Δrain+a6×ΔK+a7×year+b

where, ∆ZSD, ∆T, ∆S, ∆NO2, ∆PO4, ∆rain, ∆K are the differences from the monthly climatological values of Secchi depth, temperature, salinity, nitrite, phosphate of the ocean, precipitation at Tokyo, and mean Kuroshio position between 139°E and 140°E, respectively. The a1 to a7 were the coefficients and b was the intercept. The most suitable model was defined as the one with the lowest Akaike information criterion.

We used 10 m depth values in this study because the environments at 0 m depth may be impacted by short-term phenomena, such as rainfall and river plumes, and are inappropriate for long-term analysis. The comparison between the medians at 0 and 10 m depth was mostly similar, except for salinity (Fig. S2). The median salinity at a depth of 10 m was usually higher than that at 0 m for every subarea (Fig. S2C). In particular, the salinity at 0 m in Tokyo Bay was occasionally <25, indicating that the effects of freshwater must be considered in this subarea.

Results

Seasonality of subareas

Monthly variations in temperature at a depth of 10 m showed similar variations among the subareas (Fig. 3A); the temperature was lowest in March (February in Tokyo Bay) and highest in September. The highest temperature was not largely different (~24 °C in Tokyo Bay and the Uraga Channel, and ~25 °C in Sagami Nada and Sagami Bay), but the lowest temperature was spatially different (medians: 10.0 °C in Tokyo Bay, 12.4 °C in the Uraga Channel, 14.6 °C in Sagami Bay, and 15.1 °C in Sagami Nada).

Figure 3 Monthly variations in parameters of every subarea.

(A) Temperature, (B) salinity, (C) nitrite concentration, (D) phosphate concentration, and (E) Secchi depth in Tokyo Bay (left, TB), the Uraga Channel (second left, UC), Sagami Nada (second right, SN), and Sagami Bay (right, SB). The closed circles and vertical bars denote the monthly median and interquartile range, respectively in every subarea.

For salinity, the spatial variation was larger than the seasonal variation (Fig. 3B). In all areas, the lowest salinity was observed in September and the highest salinity was observed in March. The monthly median salinities in Tokyo Bay were 31.1–32.6, those in the Uraga Channel were 32.6–33.9, those in Sagami Bay were 33.6–34.6, and those in Sagami Nada were 33.9–34.6. In Tokyo Bay, interannual salinity variations were large in September and October (interquartile ranges: 2 and 1.6).

Variations in nitrite and phosphate levels also differed among the subareas (Figs. 3C and 3D). In Tokyo Bay and the Uraga Channel, nitrite and phosphate concentrations were not depleted ≤0.1 µM, and nitrite concentration was lowest in August, but phosphate concentration was lowest in May (Tokyo Bay) and June (the Uraga Channel). Between June and September, both nitrite and phosphate were depleted in both Sagami Nada and Sagami Bay. The phosphate concentration was highest in February in these two subareas, and the nitrite concentration was similar from December to April.

The monthly variation pattern of Secchi depth differed among the subareas (Fig. 3E). Secchi depth was deepest in January for all the subareas. In Tokyo Bay, the median Secchi depth was the shallowest, varying from 2 m (June–September) to 6 m (January). These varied from 3 m (July) to 12 m (January) in the Uraga Channel, 7 m (July) to 20 m (January) in Sagami Bay, and 12 m (May) to 23 m (January and February) in Sagami Nada.

Long-term variations

The monthly and annual rates of change in temperature at 10 m depth showed a warming trend when it was significant (Figs. 4A and 5): the threshold p-value of significance was 0.05. The annual trends based on Eq. (3) were significantly elevated in all subareas (Fig. 4A). The trends were 0.022 ± 0.0028 °C year−1, 0.014 ± 0.0029 °C year−1, 0.011 ± 0.0033 °C year−1, and 0.019 ± 0.0030 °C year−1 in Tokyo Bay, the Uraga Channel, Sagami Nada, and Sagami Bay, respectively. The detailed results of the statistical analyses (degrees of freedom, t-value, and p-value) are shown in Table S1. The monthly temperature was significantly elevated in Tokyo Bay, except during August and September (Figs. 4A and 5A). The significant monthly warming trend in Tokyo Bay with standard error was ≥0.010 ± 0.004 °C year−1, most of which were ≥0.023 °C year−1. In the Uraga Channel, Sagami Nada, and Sagami Bay, a warming trend in temperature was observed at 5, 3, and 7 months, respectively (Figs. 4A, 5B and 5C).

Figure 4 Monthly variations (J–D) of and annual (Y) changing rates of every subarea.

(A) Temperature, (B) salinity, (C) nitrite concentration, (D) phosphate concentration at 10 m depth, and (E) Secchi depth in Tokyo Bay (left, TB), the Uraga Channel (second left, UC), Sagami Nada (second right, SN) and Sagami Bay (right, SB) calculated using a linear model with a Prais-Winsten estimator. Bar height is the coefficient value, and error bar denotes 95% confidence interval. When the bar was closed, the trend was statistically significant (p < 0.05).

Figure 5 Interannual variations in median temperature at 10 m depth in (A) Tokyo Bay, (B) the Uraga Channel, (C) Sagami Nada, and (D) Sagami Bay from January (top) to December (bottom).

The closed circle and vertical bar denote the monthly median and interquartile range, respectively in every subarea. Solid line in the panel is the regression line estimated using a linear model with a Prais-Winsten estimator. In absence of the line, the interannual trend was not significant.

The annual salinity trend indicated that salinity at 10 m depth significantly (p < 0.05) decreased in Tokyo Bay (−0.0040 ± 0.0017 year−1), the Uraga Channel (−0.0048 ± 0.0013 year−1), and Sagami Nada (−0.0014 ± 0.0006 year−1), and insignificantly in Sagami Bay (−0.00091 ± 0.00090 year−1, Fig. 4B, Table S2). The significantly decreasing monthly salinity trends were temporally limited in every subarea (Figs. 4B and 6, Table S2): October in Tokyo Bay and the Uraga Channel, January and December in Sagami Nada, and November in Sagami Bay.

Figure 6 Interannual variations in median salinity at 10 m depth in (A) Tokyo Bay, (B) the Uraga Channel, (C) Sagami Nada, and (D) Sagami Bay from January (top) to December (bottom).

The closed circle and vertical bar denote the monthly median and interquartile range, respectively in every subarea. Solid grey line in the panel is the regression line estimated using a linear model with a Prais-Winsten estimator. In absence of the line, the interannual trend was not significant.

The annual trends of both nitrite and phosphate concentrations decreased significantly in all subareas (p < 0.001, Figs. 4C and 4D, Tables S3 and S4). The annual decreasing trends of these two nutrient concentrations were higher in Tokyo Bay (nitrite: −0.0077 ± 0.0024 µM year−1, phosphate: −0.0084 ± 0.0018 µM year−1) and the Uraga Channel (nitrite: −0.0081 ± 0.0020 µM year−1, phosphate: −0.0055 ± 0.0010 µM year−1), and lower in Sagami Nada (nitrite: −0.0029 ± 0.00088 µM year−1, phosphate: −0.0022 ± 0.00059 µM year−1), and Sagami Bay (nitrite: −0.0018 ± 0.00058 µM year−1, phosphate: −0.0017 ± 0.00036 µM year−1). When we see the details, the peak of nitrite concentration in Tokyo Bay and the Uraga Channel was observed around 1990, in particular, from February to September (Fig. 7). This peak was also observed from June to September at Sagami Nada and Sagami Bay (Fig. 7). However, such peaks were not observed for phosphate concentration (Fig. 8). The phosphate decreasing trends were significant from November to April in Tokyo Bay and Uraga Channel (Fig. 8). A significant decrease in phosphate was observed only in February in Sagami Nada and in February, April, August, October, and December in Sagami Bay (Fig. 8).

Figure 7 Interannual variations in median nitrite concentration at 10 m depth in (A) Tokyo Bay, (B) the Uraga Channel, (C) Sagami Nada, and (D) Sagami Bay from January (top) to December (bottom).

The closed circle and vertical bar denote the monthly median and interquartile range, respectively in every subarea. Solid grey line in the panel is the regression line estimated using a linear model with a Prais-Winsten estimator. In absence of the line, the interannual trend was not significant. The median nitrite concentrations >6 µM in Tokyo Bay and the Uraga Channel, and >2 µM in Sagami Nada, and Sagami Bay were removed from this figure.

Figure 8 Interannual variations in median phosphate concentration at 10 m depth in (A) Tokyo Bay, (B) the Uraga Channel, (C) Sagami Nada, and (D) Sagami Bay from January (top) to December (bottom).

The closed circle and vertical bar denote the monthly median and interquartile range, respectively in every subarea. Solid grey line in the panel is the regression line estimated using a linear model with a Prais-Winsten estimator. In absence of the line, the interannual trend was not significant. The median nitrite concentrations >4 µM in Tokyo Bay, >3 µM in the Uraga Channel, >2 µM in Sagami Nada, and 2 µM in Sagami Bay were removed from this figure.

The annual Secchi depth trend showed significant shallowing, except in Tokyo Bay (p < 0.05, Fig. 4E, Table S5). The annual shallowing trend was similar in Uraga Chanel (−0.076 ± 0.012 m year−1) and Sagami Nada (−0.075 ± 0.011 m year−1), and was slower in Sagami Bay (−0.046 ± 0.011 m year−1). At the monthly level, a shallowing trend was usually observed in winter (January–March, Figs. 4E and 9). In the Uraga Channel, shallow trends were observed from December to June, with the exception of May. Sagami Nada was observed from November to June, with the exception of January and March, whereas Sagami Bay was observed from January to March. In Tokyo Bay, a shallowing trend was only observed in April, while a deepening trend was observed in August with Secchi depth ≤3 m (Fig. 9A). No peak in Secchi depth was observed in the Uraga Channel, Sagami Nada, or Sagami Bay (Fig. 9).

Figure 9 Interannual variations in median Secchi depth in (A) Tokyo Bay, (B) the Uraga Channel, (C) Sagami Nada, and (D) Sagami Bay from January (top) to December (bottom).

The closed circle and vertical bar denote the monthly median and interquartile range, respectively in every subarea. Solid grey line in the panel is the regression line estimated using a linear model with a Prais-Winsten estimator. In absence of the line, the interannual trend was not significant.

The monthly variations were shown in Figs. 5–9, but the annual mean values calculated using the least squared mean technique were shown in the supplemental document (Fig. S3).

The annual precipitation amount at the Tokyo observatory was significantly increased (p = 0.0003, Fig. 10). Focusing on monthly values, the significant increasing trends were observed in August and September (p = 0.048 and 0.032, respectively, Fig. 10).

Figure 10 Monthly variations (J–D) and annual variation (Y) of changing rates in precipitation at the Tokyo Observatory calculated using a linear model with a Prais-Winsten estimator.

Bar height is the coefficient value, and error bar denotes 95% confidence interval. When the bar was closed, the trend was statistically significant (p < 0.05). The monthly variation was calculated using the monthly integrated precipitation and the annual variation was using the yearly integrated one.

The nitrate plus nitrite concentration in Sagami River water was significantly increased from 1970 to 2020 (coefficient ± SE: 0.021755 ± 0.005785, p = 0.000425), while the peak was observed 1990–2000 and gradually decreased after 2000 (Fig. 11A). In fact, nitrate plus nitrite concentration in Sagami River water significantly decreased after 1995 (coefficient ± SE: −0.006199 ± 0.001120, p < 10−5). The TN and TP concentrations were significantly decreased (p < 0.005). As well as nitrate plus nitrite concentration, the TN and TP gradually decreased after 2000 (Figs. 11B and 11C).

Figure 11 Interannual variations in (A) nitrate plus nitrite, (B) total nitrogen (TN), and (C) total phosphate (TP) concentrations in Sagami River.

Solid line in the panel is the regression line estimated using a linear model with a Prais-Winsten estimator.

Contributions of Secchi depth anomaly

Based on Eq. (4) and the AIC values, the contributions of temperature, salinity, nitrite, phosphate, and Kuroshio axis position to the Secchi depth anomaly variation were estimated throughout the year (Fig. 12, Table S6). The precipitation anomaly at Tokyo has never been selected in all the subareas. In Tokyo Bay, only salinity anomalies remained in the final (least-AIC) model to explain the Secchi depth anomaly, and the relationship between the Secchi depth and salinity anomalies was significantly positive (i.e., an increase in salinity deepened the Secchi depth). This significant positive relationship between Secchi depth and salinity anomalies was the same as that in the other subareas (Fig. 12B). In the Uraga Channel, temperature, salinity, and phosphate and Kuroshio axis anomalies remained in the final model, and the coefficient indicated that the Secchi depth increased with increases in the temperature and salinity anomalies (Fig. 12). The effects of phosphate and Kuroshio axis on the Uraga Channel remained in the least-AIC model, but insignificant (p > 0.07). In the Sagami Nada, the variables of the full model except precipitation and Kuroshio axis anomaly were selected. Elevations in temperature were related to the deepening of the Secchi, but the effects of phosphate were not significant. In Sagami Bay, this trend was not selected as the final model, and warming and phosphate decrease deepened the Secchi depth; the coefficient of the nitrite anomaly was found to be insignificant (p > 0.05).

Figure 12 Coefficients to the Secchi depth anomaly.

(A) Temperature, (B) salinity, (C) nitrite, (D) phosphate, (E) Kuroshio axis, and (F) year in Tokyo Bay (TB), the Uraga Channel (UC), Sagami Nada (SN), and Sagami Bay (SB) in a linear model with a Prais-Winsten estimator. The bar height indicates coefficient value, and bar denotes the 95% confidence interval. NS shows that the parameter was not selected in the least-AIC model.

Since the Secchi depth of Tokyo Bay and Uraga Channel was several meters, the 10 m depth was not in the euphotic layer, 1% of surface radiance, and the contributions of parameters would be different when we used the 0 m variables. However, we noted that when we used the 0 m variables instead of the 10 m depth variables in Tokyo Bay, the phosphate anomaly was selected in the final model as well as the salinity anomaly, but it was insignificant (p = 0.319). The coefficient of salinity anomaly was significantly positive (0.279 ± 0.117, p = 0.017), which was similar to that when using 10 m depth values. In Uraga Channel, the Kuroshio axis was not in the final model compared to that using 10 m depth values, and the other remaining explanatory variables were the same: the relationships with temperature (0.6585 ± 0.1373, p < 10−5) and salinity (2.3044 ± 0.2829, p < 10−20) were significantly positive as well as that using 10 m depth values, and that of phosphate was insignificant (0.4816 ± 0.2781, p = 0.084).

Discussion

This study was aimed to identify possible trends in coastal ocean productivity in the western part of the Great Tokyo area based on Secchi depth variations. The observation area was divided in four subareas based on temperature and salinity, and significant Secchi-depth-shallowing trends were detected in all subareas, except for Tokyo Bay. Secchi depth has been shown to be a suitable index of chlorophyll a concentration (Mackas, 2011). According to Nixon (1995) definition of eutrophication, “an increase in the rate of supply of organic matter to an ecosystem,” shallowing trends of Secchi depth indicated that eutrophication occurred in the coastal area of Great Tokyo, while the sea surface temperature was warming, and nitrite and phosphate concentrations were sometimes decreasing. This eutrophic trend was different from that in the Seto Inland Sea (Abo & Yamamoto, 2019; Yamamoto, 2003) and the Mediterranean Sea (Derolez et al., 2020a; Derolez et al., 2020b) where oligotrophication has been reported.

A shallowing trend of Secchi depth was observed in the Uraga Channel (outer Tokyo Bay), Sagami Nada, and Sagami Bay. The long-term trend of primary productivity in the considered subareas is reported for the first time in this study. Along with the eutrophication trends, we detected a warming trend in temperature, a freshening trend in salinity, and a decreasing trend in nitrite and phosphate concentrations in the annual mean values (Fig. 4). These results suggest that warming and freshening of the surface water induce the development of a pycnocline and improve light conditions, and phytoplankton growth occurs with nutrient consumption. This hypothesis was reasonable, particularly during winter as the Secchi depth in winter was the deepest in every subarea and nutrient concentration was replete at that time (Fig. 3), indicating that light limitation generally occurred in winter along with significant eutrophication trends being observed during winter (January to March). In addition, the light limitation for primary production in winter has been previously reported in Sagami Bay (Ara et al., 2011).

However, relationships between the Secchi depth anomaly and other parameters showed different patterns, particularly between the temperatures (Fig. 12). The linear model approach indicated that the Secchi depth deepened in the warming water (Fig. 12), while long-term Secchi depth shallowing occurred with the warming (Fig. 4). We also conducted the same linear model approach in the winter datasets (January–March); although the details of the final models were different, the positive effects of temperature (Secchi depth deepened when the temperature was elevated) were similar in the Uraga Channel, Sagami Nada, and Sagami Bay (Fig. S4). The approach in Eq. (4) was not only focused on the long-term variations, but can explain more short-term variations of Secchi depth as well as the long-term variations. Therefore, the inconsistent results between the two approaches indicate not only an improvement of the photoenvironment for phytoplankton growth but also other changes such as nutrients and salinity contribute to the Secchi depth variations in our observation area.

First, local riverine flow significantly contributes to eutrophication in the coastal area of Sagami Bay (Ara & Hiromi, 2008). Nitrogenous nutrient concentration (nitrate plus nitrite) in Sagami River increased to approximately 1995 and then decreased as like TN concentration (Figs. 11A and 11B). Unfortunately, we had to focus on the nitrite concentration, which may not be a good indicator of nitrogenous nutrient concentrations; however, its concentration sometimes peaked around 1990 (Fig. 7). However, this increase in nitrite in the coastal area was not associated with shallowing of the Secchi depth (Figs. 7, 9, and 12), and the Secchi depth was not clearly deepened (i.e., oligotrophication occurred) after 2000 (Fig. 9). The phosphate inputs suggested by TP concentration also declined after the 1990s (Fig. 11C), and it was matched with the decline of phosphate concentration in the coastal area (Fig. 8). However, the decline of phosphate concentration was associated with the deepening the Secchi depth in Sagami Bay (Fig. 12), which was not observed (Fig. 9). Therefore, although river-origin nutrients are vital for primary production in Sagami Bay (Ara et al., 2011) and our nitrite concentrations supported this, we concluded that river inputs might not have a severe impact in the considered area.

Second, horizontal advective transport of eutrophic seawater (i.e., Tokyo Bay water) can also contribute to the eutrophication trend. Because Tokyo Bay water frequently overflows into Sagami Bay and Sagami Nada (Aoki et al., 2022; Furushima, 1996; Yanagi & Hinata, 2004), water from Tokyo Bay could contribute to the eutrophication of these subareas because of water mixing. In the Great Tokyo area, the precipitation was increasing (Fig. 10) while their anomaly was not contributed to the Secchi depth variations (Fig. 12). In addition, the import of fresh water from the neighboring catchment area (i.e., some water will possobly flow into Sagami Bay is imported and flows into Tokyo Bay) was also increased, and thus the water residence time of Tokyo Bay in 2002 was shortened by two-thirds of those averaged from 1947 to 1974 (Okada et al., 2007). These suggested that horizontal advection was activated in these 60 years, while Aoki et al. (2022) suggested that the middle-term (from 2010 to 2019) activation of horizontal advective transport from Tokyo Bay did not prospect. Fewer saline, cold, and eutrophic waters were observed in Tokyo Bay, particularly during the winter (Fig. 3). Thus, the occurrence of eutrophication should involve lowering of the temperature and salinity, and the observed relationships between the Secchi depth anomaly and other parameters (Fig. 12) likely support horizontal advective transport from Tokyo Bay contribute to the eutrophication. However, the warming trend was significant in winter with the eutrophication trend (Fig. 4). In addition, both nitrite and phosphate concentrations were high in Tokyo Bay (Figs. 7 and 8), but these nutrients usually showed decreasing trends (Fig. 4). Therefore, we attribute that horizontal advective transport is not the main reason for the long-term eutrophication trend. However, in some months, the decrease in Secchi depth was not associated with the warming (Fig. 4). We have no evidence and the cooling and nutrient-increasing trends were not observed, but horizontal advection may contribute to the eutrophication trend in such months.

Third, the decline of horizontal advective transport of oligotrophic Kuroshio seawater could contribute to the eutrophication, because the Kuroshio intrusion to Sagami Bay was usually observed and the Kuroshio path contributed to the intrusion pattern (Hinata et al., 2005; Matsuyama et al., 1999). However, the Kuroshio path was not contributed to the Secchi depth variation in our study (Fig. 12). In the Japanese coastal area, the Kuroshio path was usually assumed as the explanatory variable of Secchi depth, but clear relationships were not detected (Hagiwara, 2009; Ichikawa, Yamamoto & Hirota, 2009) as well as in our present study. In our study, however, we only compared the Kuroshio axis position off Sagami Bay (139–140°E), and the Kuroshio intrusion also impacts the temperature and salinity values which were included in the models, suggesting there might be a better approach to identify the effect of Kuroshio. Thus, we cannot conclude that the Kuroshio axis position is not important for the Secchi depth anomaly in the coastal area of Kanagawa, but it was not the main cause of Secchi depth variations.

On the basis of the above discussion, the inconsistent results of the correlation analyses (Fig. 12) to the long-term variation of Secchi depth (Fig. 4) indicated the difference in temporal scale. The Secchi depth could vary with horizontal advective transport of riverine water, Tokyo Bay water, and Kuroshio water, but the effects of horizontal advective transport were not clearly detected in the trend analysis. Therefore, the long-term variation in Secchi depth may have been caused mainly by a better photoenvironment for phytoplankton growth.

Here, the warming of seawater was important for the eutrophication trend. One of the heat sources of the coastal area is urban waste heat, such as warm wastewater. In the Great Tokyo area, Kinouchi, Yagi & Miyamoto (2007) report that the urban waste heat flux is several hundred TJ per day, and it contributed to warming the stream water temperature. If we assume that the urban heat flux is 4.2 × 102 TJ per day, then the urban heat increases the temperature of 107 L of water by 0.01 °C per day. This is based on the assumption that it takes 4.2 J to increase the temperature of 1 g of water by 1 °C. The warming trend was observed at a depth of 10 m in our study. This means that the urban heat elevates the temperature of an area of 103 m2 by 0.01 °C day−1, and an area of 3.7 × 105 m2 by 0.01 °C year−1. Tokyo Bay is ~1.5 × 109 m2, so the urban waste heat flux cannot sufficiently explain the increasing trend of seawater temperature in our study. In Kinouchi, Yagi & Miyamoto (2007), the warming trend of stream water was quite low in the downstream area, supporting the idea that the urban waste heat contribution to warming seawater is quite limited.

Therefore, it was suggested that the warming trend which was detected in our study is mostly due to global warming, and global warming contributes to eutrophication in the coastal region of the Great Tokyo area.

Tokyo Bay (inner Tokyo Bay) was the only subarea where we could not detect any eutrophication/oligotrophication trends based on annual mean Secchi depth values. Previous studies have reported that nutrient concentrations decrease environmental pollution is improving (Ando et al., 2021; Kubo et al., 2019), thus, the Secchi depth is deepening in Tokyo Bay (Ishii, 2009). Our results for Secchi depth in Tokyo Bay were inconsistent with those of Ishii (2009), whereas the nutrient (phosphate) decline was consistent with previous studies (Ando et al., 2021; Kubo et al., 2019). We considered the inconsistency of Secchi depth to be due to the low sensitivity in our dataset; the depth was recorded at a 1-m pitch, whereas the Secchi depth was 1–2 m in inner Tokyo Bay; thus, our resolution was insufficient to identify the long-term changes in Tokyo Bay. The other possible reason was the spatial difference. The previous studies were not observed along the Kanagawa coastal area, and the west (Kanagawa)-east (Chiba) gradient of water chemistry was reported in Tokyo Bay (Ando et al., 2021). The current structure, at least the tidal current, was different between the west and east sides (Hosokawa & Okura, 2022), and thus the spatial variations were also expected and must be evaluated in the future.

Conclusions

In this study, we evaluated the trend of primary production in the coastal ocean of the megacity Great Tokyo based on datasets over a half-century. Temperature increase, and eutrophication were observed in the study area. We considered that the short-term trend in primary production was attributable to horizontal advection from Tokyo Bay, particularly in winter, whereas the development of a shallower pycnocline due to warm temperatures improved the photoenvironment in the ocean, leading to long-term eutrophication. Previous studies have indicated that primary production has largely decreased in the western North Pacific (Boyce, Lewis & Worm, 2010), our study revealed differences at a spatial scale and showed that the ocean is more complex, particularly in coastal areas. Even downstream of the megacity, eutrophication and oligotrophication cannot be simplified.

In the present study, we first hypothesized that the decline in fish production in Sagami Bay (Takamura, Katayama & Kinoshita, 2016) occurred due to oligotrophication. However, it was found that eutrophication occurred in Sagami Bay and Sagami Nada, suggesting that the “oligotrophication hypothesis” can be rejected. Nixon et al. (2009) reported that fishery production decreases with warming. We cannot reject the possibility that temperature elevation (Figs. 4 and 5) decreased fish production. This suggests that adequate fishery stock management is strongly recommended to improve fishery production in this area under global warming conditions based on the long succession of oceangoing datasets, rather than practical approaches to increase nutrient concentrations such as sediment plowing and relaxing the discharge of treated sewage in the Seto Inland Sea (Abo & Yamamoto, 2019).

Supplemental Information

Supplemental Information 1 Relationship between Forel-Ule scale water color and Secchi depth.

The color indicated the number of observations.

Click here for additional data file.

Supplemental Information 2 Relationships of parameters between 0 m and 10 m depths.

(a) temperature at the surface and 10 m depth, (b) salinity at the surface and 10 m depth, (c) nitrite concentration at the surface and 10 m depth, and (d) phosphate concentration at the surface and 10 m depth in Tokyo Bay (left), Uraga Channel (second left), Sagami Nada (second right), and Sagami Bay (right). The dotted line is 1:1 line.

Click here for additional data file.

Supplemental Information 3 Interannual variations in the least squared mean temperature, salinity, nitrite concentration, phosphate concentration, and Secchi depth in every subarea.

(a) Tokyo Bay, (b) the Uraga Channel, (c) Sagami Nada, and (d) Sagami Bay. The parameters except Secchi depth was collected at 10 m depth. The closed circle and vertical bar denote the yearly least squared mean and standard errors, respectively in every subarea. The regression line was not shown. The least squared mean values were calculated from a model whose description was a following equation: parameters ~ lm(f.year + f.month). Here, f.year and f.month denote year transformed as a categorical value, and month transformed as a categorical value, respectively. The “parameter” were median values of temperature, salinity, nitrite concentration, phosphate concentration, and Secchi depth in every observation of every subarea.

Click here for additional data file.

Supplemental Information 4 Coefficients to the Secchi depth anomaly in winter (January–March).

(a) temperature, (b) salinity, (c) nitrite, (d) phosphate, and (e) year in Tokyo Bay (TB), the Uraga Channel (UC), Sagami Nada (SN), and Sagami Bay (SB) in a linear model with a Prais-Winsten estimator. The bar height is the coefficient value, and the bar denotes the 95% confidence interval. NS shows that the parameter was not selected in the least-AIC model.

Click here for additional data file.

Supplemental Information 5 The statistical results of time-series analysis of temperature.

SE and DF denote standard errors and degree of freedom, respectively.

Click here for additional data file.

Supplemental Information 6 The statistical results of time-series analysis of salinity.

SE and DF denote standard errors and degree of freedom, respectively.

Click here for additional data file.

Supplemental Information 7 The statistical results of time-series analysis of nitrite concentration.

SE and DF denote standard errors and degree of freedom, respectively.

Click here for additional data file.

Supplemental Information 8 The statistical results of time-series analysis of phosphate concentration.

SE and DF denote standard errors and degree of freedom, respectively.

Click here for additional data file.

Supplemental Information 9 The statistical results of time-series analysis of Secchi depth.

SE and DF denote standard errors and degree of freedom, respectively.

Click here for additional data file.

Supplemental Information 10 The statistical results of correlated analyses of Eq (4).

∆T, ∆S, ∆NO2, ∆PO4, and ∆K denote the difference from monthly median values of temperature, salinity, nitrite concentration, phosphate concentration, and Kuroshio axis position, respectively.

Click here for additional data file.

Supplemental Information 11 Data of Kanagawa Prefecture for Akada Kodama and Yamaguchi.

“NA” denote the data was not available.

Click here for additional data file.

The present dataset is a product of the enormous efforts of many researchers, officers, crews, and operators at the Kanagawa Prefectural Fisheries Technology Center. The valuable comments for the early manuscript were provided by Dr. Aoki, Fisheries Resources Institute. The open-source data were downloaded from the Japan Meteorological Agency website and the water information system website. We also appreciate Dr. Kubo and another anonymous reviewer.

Additional Information and Declarations

Competing Interests

Author Contributions

Data Availability

The authors declare that they have no competing interests.

Hideyuki Akada conceived and designed the experiments, performed the experiments, analyzed the data, prepared figures and/or tables, authored or reviewed drafts of the article, and approved the final draft.

Taketoshi Kodama conceived and designed the experiments, performed the experiments, analyzed the data, prepared figures and/or tables, authored or reviewed drafts of the article, and approved the final draft.

Tamaha Yamaguchi conceived and designed the experiments, performed the experiments, analyzed the data, prepared figures and/or tables, authored or reviewed drafts of the article, and approved the final draft.

The following information was supplied regarding data availability:

The data is available in the Supplemental File and at the Sea condition survey project result report: https://www.pref.kanagawa.jp/docs/mx7/cnt/f430694/p789241.html.

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
