# Peer review of "Eutrophication trends in the coastal region of the Great Tokyo area based on long-term trends of Secchi depth"

_PeerJ, doi:10.7717/peerj.15764_

## Round 0.1 · original submission · Minor Revisions

The general comments from the two reviewers were positive and I agree with them.

·

Basic reporting

The manuscript by Akada, Kodama, and Yamaguchi involves a precious data set of Secchi depth derived from hard work, compiling data from continuous observations, and assessing the details of each area of Tokyo Bay and Sagami Bay. The content is also apparent; therefore, this paper is worth publishing in PeerJ with minor modifications.

Can increasing water temperatures be interpreted as an effect of global warming? To what extent does urban waste heat (e.g., from sewage treatment plants and power plant cooling water) have an impact? The manuscript described the water temperature increase as "warming" but partly as "global warming."

Is it not necessary to consider the meandering of the Kuroshio Current? And if necessary, is it reasonable to analyze the data for the entire period?

Line 102-104 The conclusion of Aoki et al. (2022) also suggests that the main factor of the oligotrophication in Tokyo Bay is the decrease in artificial loads. Therefore, writing them in parallel as two main factors can greatly mislead the reader.

Experimental design

No comment.

Validity of the findings

No comment.

Additional comments

No comment.

Reviewer 2 ·

Basic reporting

.

Experimental design

.

Validity of the findings

.

Additional comments

See attachment.

Annotated reviews are not available for download in order to protect the identity of reviewers who chose to remain anonymous.

---

## Round 0.2 · accepted · Accept

Congratulations! Now your munuscript is ready for publication.

·

Basic reporting

I have read the revised manuscript, reviewers' comments, and the authors' responses. I found the comments were appropriately dealt with. I have no further comments.

Experimental design

no comment

Validity of the findings

no comment

Additional comments

no comment

Reviewer 2 ·

Basic reporting

.

Experimental design

.

Validity of the findings

.

Additional comments

Review of Revised version of PeerJ #85339
Eutrophication trends in the coastal region of the Great Tokyo area based on long-term trends of Secchi depth

The authors analyzed extensive data sets on water quality collected in Great Tokyo area to discuss the long-term trends of the Secchi depth. The manuscript of the background and methodology of this study is clearly written, and reasonable results are obtained. Additionally, the discussions provide important/valuable information for general readers of PeerJ.
The reviewers’ comments and suggestions are well incorporated in the revised version of the manuscript. Some points that have not been incorporated are reasonably explained by authors’ reply.

Just one thing the reviewer would like to point out is the journal name of Okada et al., 2007. The Japan Society of Civil Engineers recommend to use the English journal title as:
Journal of Japan Society of Civil Engineers, Ser. B
So, ‘Doboku Gakkai Ronbunshuu B’ in the References (Line 576-577) can be replaced the above.

I commend the authors for their extensive data set, compiled over many years of detailed fieldwork. The manuscript is clearly written in professional, unambiguous language. There is no apparent weakness. The reviewer carefully checked the text to find it should be accepted as it is.